# How Well Do Semen Analysis Parameters Correlate with Sperm DNA Fragmentation? A Retrospective Study from 2567 Semen Samples Analyzed by the Halosperm Test

**DOI:** 10.3390/jpm13030518

**Published:** 2023-03-13

**Authors:** Shiao Chuan Chua, Steven John Yovich, Peter Michael Hinchliffe, John Lui Yovich

**Affiliations:** 1PIVET Medical Centre, Perth, WA 6007, Australia; 2Hospital Shah Alam, Shah Alam 40000, Selangor, Malaysia; 3Curtin Medical School, Curtin University, Perth, WA 6102, Australia

**Keywords:** sperm DNA fragmentation (SDF), Halosperm test, sperm DNA fragmentation index (DFI), in vitro fertilization (IVF), intracytoplasmic sperm injection (ICSI)

## Abstract

Sperm DNA fragmentation (SDF) levels have been measured in the workup for in vitro fertilization (IVF) at PIVET since 2007, with the Halosperm test having replaced the previous sperm chromatin structure assay (SCSA) since 2013. Of 2624 semen samples analyzed for the Halosperm test, 57 were excluded as the sperm concentration was <5 million/mL, a level too low for accurate testing, leaving 2567 samples for assessment within this study. The SDF rates were categorized in 5 sperm DNA fragmentation indices (DFI), ranging from <5% to levels >30%, and these categories were correlated with the respective semen analysis profiles and two clinical parameters, namely the age of the male and the ejaculatory abstinence period prior to the sample. The results showed a significant correlation with male age (r = 0.088; *p* < 0.0001), the abstinence period (r = 0.076; *p* = 0.0001), and the semen volume (r 0.063; *p* = 0.001), meaning an adversely high SDF was associated with advanced age, prolonged abstinence, and raised semen volume parameters. There was a significant negative correlation with sperm morphology (r = −0.074; *p* = 0.0001), progressive motility (r = −0.257; *p* < 0.0001), and semen pH (r = −0.066; *p* < 0.001), meaning these semen anomalies were associated with high SDF values. With respect to abnormal morphology, sperm tail defects had a positive correlation (r = 0.096; *p* < 0.0001) while midpiece defects showed a negative correlation (r = −0.057; *p* = 0.004), meaning that tail defects are most likely to associate with adverse DFI values. With respect to motility patterns, the poorer patterns showed a positive correlation with increased DFI, namely C pattern (r = 0.055; *p* = 0.005) and D pattern (r = 0.253; *p* < 0.0001). These results imply that raised DFI reflects poor sperm quality and should be investigated in clinical trials involving IVF and the consideration of intracytoplasmic sperm injection (ICSI).

## 1. Introduction

The World Health Organization (WHO) defines infertility as a disease of the male or female reproductive system when a couple fails to conceive after 12 months or more of regular unprotected sexual intercourse [1]. Infertility problems are on the rise and approximately 8–12% of couples are affected in the reproductive age group worldwide [2]. Of all infertility cases, male factor contributes approximately 50% of the causation among heterosexual couples, and the majority of these cases have unexplained aetiology [3,4,5].

Data on semen quality collected systematically from reports published worldwide indicate that spermatozoa density has declined appreciably across the years 1938–1990 [6]. In the 1940s, the consensus of opinion was that a volume of less than 1–1.5 mL after an ejaculatory pause of a couple of days or more is abnormal [7]. However, semen volume of 1.4 mL has been on the fifth percentile and is considered as a normal reference according to WHO 6th edition 2021 [8]. The lower reference value for a “normal” sperm count has also changed from 60 × 10^6^/mL in the 1940s [7] to the present value of 16 × 10^6^/mL [8].

Some authors demonstrated that pregnancy outcomes following in vitro fertilization (IVF) and intracytoplasmic sperm injection (ICSI) are not related to sperm DNA fragmentation index (DFI), and the finding of elevated sperm DFI does not impact oocyte fertilization or embryo development [9]. However, other authors found that sperm DNA damage contributes to a negative predictive factor for couples undergoing assisted reproductive technology (ART) [10]. Nonetheless, despite the demonstration of high levels of sperm DNA fragmentation (SDF) in the sperm sample, IVF may occur normally, particularly when the ICSI technique is applied, and SDF may be corrected by the repair capability of oocytes from younger females, therefore not affecting optimal embryo development [11].

Although the DFI was not strongly correlated with conventional semen parameters, it can be regarded as having an adjunctive role in male infertility investigation, providing an additional predictive factor for fertilization and pregnancy outcome in couples undergoing ART, since it is also a sensitive molecular test which investigates the paternal genome [12,13,14]. In addition, an SDF integrity assay could provide a promising biomarker for clinical andrology [15,16].

A normospermic semen sample, including sperm concentration, morphology, and motility, does not always ensure normal sperm DNA integrity. One study applying the sperm chromatin structure assay (SCSA) in men with normal standard semen parameters showed the Odds Ratio (OR) for infertility was increased with DFI levels >20%, and it further increased if one of the semen parameters was abnormal [17]. This study accorded with earlier SCSA studies at PIVET reported in 2007 at the 26th Annual meeting of the Fertility Society of Australia [18]. Subsequently, after a comparative trial showed similar DFI ratings, the more cost-effective Halosperm test replaced SCSA from 2013. Therefore, the objective of this study is to explore the association between sperm quality assessed by conventional semen analysis and sperm DNA integrity assay by applying the Halosperm test.

## 2. Materials and Methods

### 2.1. Study Design

This retrospective study period was from 1 March 2013 until 30 September 2022. Men were included into the study if they were undertaking semen analysis and sperm chromatin dispersion (SCD) test using the Halosperm test (halosperm^®^ G2 by halotech^®^ DNA) at PIVET Medical Centre. Exclusion criteria for this study included azoospermia and sperm concentration less than 5 × 10^6^/mL due to the technical difficulty of interpreting the normal morphology rate and diluting to a working concentration of ≤20 × 10^6^/mL for preparation of the Halosperm test.

The working dataset was extracted from our extensive database using Filemaker Pro 12 (Apple, Cupertino, CA, USA), which undergoes regular validation checks. A flow diagram of data extraction is shown in Figure 1.

### 2.2. Laboratory Procedures

Semen samples were collected and analyzed following the WHO standard 2010 [15], with macroscopic examination including the liquefaction, semen viscosity, volume, and pH. Microscopic examination assessed the sperm concentration, morphology, and motility. The interpretation of results was in accordance with WHO 5th edition.

#### 2.2.1. Semen Analysis


*Terminology*


Throughout this manuscript we have applied the terms sperm and spermatozoa interchangeably to optimize readability.


*Liquefaction*


Semen is typically a semi-solid coagulum following immediate ejaculation, which functions to protect the sperm from the hostile acidic vaginal environment. At room temperature, complete ejaculate liquefaction is normally achieved within 15–30 min, and sperm gain the ability to move. Normal liquefied semen samples may contain gelatinous bodies which do not liquefy; these do not appear to have any clinical significance or adverse relevance. The presence of mucus strands may interfere with semen analysis [15].


*Semen Viscosity*


The viscosity of the sample reflects the mucous content. High viscosity can interfere with the physical and chemical characteristics of seminal fluid [19]. High viscosity may reflect past or present genital tract infection or infrequent ejaculation.


*Semen Volume*


Two-thirds of the ejaculate volume is contributed by the seminal vesicles, one-third is from the prostate gland, with a small amount from the bulbourethral glands, testicle, and epididymis [20]. Various factors such as the method and timing of collection can affect its volume. The latest WHO lower reference value for semen volume is 1.4 mL [8].


*Semen pH*


The pH of semen reflects the balance between the pH values of the alkaline seminal vesicle fluid and the acidic prostatic fluid. Semen pH below 7.2 indicates a lack of alkaline seminal vesicle fluid [8].


*Sperm morphology*


Sperm morphology is a key element for both sperm motility and the binding of the sperm to the zona pellucida. More recent studies suggest that sperm with “normal” morphology have reduced levels of sperm DNA damage [21,22,23,24]. It has also been noted that sperm displaying a tight tail-curling response to hypo-osmolar stress have low levels of DNA fragmentation along with the features of normal morphology [25]. The lower reference value for semen morphology has not been changed since 2010, i.e., at 4% [8].

The smear of the sperm was air dried and then stained by Diff-Quick. The morphology of sperm head shape and size, sperm neck and midpiece, tail, and cytoplasmic droplets were determined under brightfield optics at 1000× magnification with oil immersion. At least 200 sperms were counted to calculate the percentage of both normal and abnormal morphology.


*Sperm concentration*


Sperm concentration is highly variable between ejaculates and between men. The actual count, unless low, does not reflect the fertility of a sample since pregnancies arise in fertile populations over a wide range of sperm concentrations. Sperm were examined on the haemocytometer with phase contrast optics at 200× magnification. The lower reference value is 16 × 10^6^/mL [8].


*Sperm motility*


Sperm motility is required for the sperm to migrate from the seminal plasma into cervical mucus and then forward into the uterine and tubal environment. As sperm migrate out of the cervical mucus, they undergo a process of capacitation that is the precursor to the development of fertilisation potential.

Motility is provided by the action of the flagellum moving in a circular corkscrew motion and is sensitive to the health of the spermatozoa. Motility is a function of both energy driven tail motion and the environment in which the sperm is swimming. Therefore, the rate of motility will vary between that in seminal plasma and washing media after sperm separation processes. Motility also decreases with time and lower temperature. The lower reference value for total motility has been changed from 40% to 42% and progressive motility has been changed from 32% to 30% [8,15].

At PIVET, we classify motility into four categories:Rapid progressive motility (A-grade sperm)—movement approximately equal to five head lengths per second.Slow or sluggish progressive motility (B-grade sperm).Non-progressive motility (C-grade sperm).Non-motile (D-grade sperm).

Progressive motility count embraces the categories of A- and B-grade sperm.

#### 2.2.2. Sperm Chromatin Dispersion Test—Halosperm^®^ G2

The DNA in sperm is repackaged to condense the chromosomes into a tight bundle to facilitate sperm form and function. The repackaging is managed by the replacement of histones with transitional proteins and protamines, which act to protect the DNA from external stress [26]. The main stress is from oxidative damage derived either from internal metabolism or external sources such as leucocytes. Oxidative damage has been linked to fragmented DNA, poorer embryology development, and increased rates of miscarriage [27,28].

In recent years, IVF clinics have been testing sperm DNA Damage rates using a variety of tests, one of which is the Halosperm test. Other popular tests are Sperm Chromatin Structure Assay (SCSA) and terminal deoxynucleotidyl transferase dUTP nick end labelling (TUNEL). PIVET has elected to use Halosperm as the preferred routine test to be conducted in house. PIVET introduced Halosperm in early 2013, replacing SCSA as the primary sperm function test outside semen analysis, which in turn was a replacement from the earlier Acrosome Response to ionophore Challenge (ARIC) test [22].

##### Principle of the Halosperm Test

Intact unfixed sperm (fresh, frozen, or unthawed, diluted samples) were immersed in an inert agarose microgel on a pre-treated slide. An initial acid treatment denatures the DNA in fragmented DNA sperm cells. Thereafter, the lysis solution was applied to remove most of the nuclear proteins. Sperm heads with massive loops of elongated DNA strands emerging from the central core signifies the absence of DNA breakage and show a dispersion of large to medium halo. The nucleoids from sperm with fragmented DNA either do not show a dispersion halo or show a small halo. Figure 2 depicts the varying degree of halo dispersion and without halo.

##### Procedure

Sperm DNA fragmentation (SDF) was performed using the Halosperm^®^ G2 kit (Parque Cientifico de Madrid, Madrid, Spain). Halosperm cannot be performed on neat sperm concentrations of <5 × 10^6^ million/mL. Samples with counts between 5–19 million/mL should be concentrated by centrifuging at 2000 rpm for 10 min to obtain a suitable concentration for Halosperm testing.

The neat sperm sample was diluted in isotonic sodium chloride buffer to a maximum concentration of 20 million sperm per mL. 50 µL of each diluted semen aliquot was mixed with 100 µL of the melted agarose gel and placed in the 37 °C water bath to prevent gelification.

8 µL of the sperm–agarose mixture was added to the labelled super-coated slides and covered with coverslips at room temperature (22 °C). The slides were then transferred into a refrigerator at 4 °C for 5 min.

Post-refrigeration, an initial acid treatment, followed by a lysis solution treatment, was applied, and washing, dehydration, and multicolored staining of the slides was performed at room temperature.

#### 2.2.3. Assessment under Light Microscopy

The protocol at PIVET Medical Centre requires a total of 200 sperm cells to be assessed under bright field microscopy. Those spermatozoa observed with a large halo, where the halo width was similar or larger than the diameter of the core, were classified as spermatozoa without any DNA fragmentation. Similarly, those spermatozoa with a medium-sized halo were also classified as having no DNA fragmentation.

Those spermatozoa observed with a halo width similar or smaller than one-third of the diameter of the core were classified as spermatozoa with DNA fragmentation. Similarly, those spermatozoa without a halo and spermatozoa, showing features of degradation where the core was irregularly or weakly stained, were included as spermatozoa with DNA fragmentation. However, those cells which did not exhibit a clear tail were not included in the sperm count for DNA fragmentation.

### 2.3. DNA Fragmentation Index (DFI) Calculation

The percentage of sperm with fragmented DNA:$$DFI\;\left(\%\right)=\frac{Fragmented\;and\;degraded}{Total\;number\;of\;cells\;counted}\times 100$$

PIVET Medical Centre DFI cutoffs and interpretation:0–4.9%: Excellent sperm DNA Integrity5–14.9%: Adequate sperm DNA Integrity15–29.9%: Elevated levels of DNA fragmentation. This may impact upon fertility potential; ICSI recommended.≥30%: Severely elevated levels of DNA fragmentation. This is very likely to impact upon fertility potential. ICSI required.

The Halosperm test has quality control assessment from FertAid Pty Ltd. (Largs, Australia), which provides quality assurance and training in the Reproductive Sciences, and it monitors many aspects of the embryology and andrology within PIVET laboratory.

### 2.4. Statistical Analysis

We performed all statistical analyses using SPSS software (version 26.0, SPSS Inc., Chicago, IL, USA). All numeric data are presented as the mean value ± standard deviation (SD), by One way ANOVA test. Correlation analysis was performed using bivariate Pearson analysis. Multivariate logistic regression was applied to determine confounding variables. Differences between the values were considered statistically significant when *p* < 0.05.

## 3. Results

### 3.1. Patient Characteristics

A total of 2624 men who had undergone the Halosperm test from 1 March 2013 until 30 September 2022 were retrieved from Filemaker Pro 12 (Apple, USA). Of these men, 57 were excluded from this study as their sperm concentrations were below 5 × 10^6^/mL. Essentially, 2567 men were included in this study.

The DFI ratings were categorized into 5 groups, namely group 1 (DFI < 5%), 2 (DFI 5.1–10%), 3 (DFI 10.1–15%), 4 (DFI 15.1–29.9%), and 5 (≥30%). There were 548 patients in group 1 (21.3%), 933 patients in group 2 (36.3%), 477 patients in group 3 (18.6%), 470 patients in group 4 (18.3%), and 139 patients in group 5 (5.4%). Table 1 showed the semen and clinical characteristics in each of the 5 groups.

Within each of the DFI groups, statistical analysis is also reported. Figure 3 showed that mean DFI for the respective groups was 3.36% ± 1.16% (95% CI: 3.27–3.46%) for Group 1; 7.15% ± 1.38% (95% CI: 7.06–7.24%) for Group 2; 12.03% ± 1.37 (95% CI: 11.91–12.15%) for Group 3; 20.12% ± 4.04% (95% CI: 19.76–20.49%) for Group 4; and 43.78% ± 14.36% (95% CI: 41.38–46.20%) for Group 5, respectively.

### 3.2. Correlations between Routine Semen Parameters and Sperm DFI

Table 2 shows the summary of correlation coefficients between sperm DFI and semen parameters. Correlation analysis showed that sperm DFI was negatively associated with normal morphology, total motility, progressive motility, and pH (correlation coefficients r was −0.074, −0.257, and −0.066; *p* = 0.0001, *p* < 0.0001, *p* < 0.001, and *p* < 0.001, respectively).

DNA is known to be held in a tightly coiled and packaged form in the head of the sperm. Surprisingly, in our study we found that there was no correlation between sperm displaying head defects and sperm DFI (r = 0.009; *p* = 0.645). On the other hand, we found that sperm displaying tail defects was positively correlated with sperm DFI (r = 0.096; *p* < 0.0001), while sperm displaying midpiece defects was negatively correlated with sperm DFI (r = −0.057; *p* = 0.004). In other words, this means that the lower the proportion of tail defects, the lower is the DFI; this appears to be the stronger predictor for the sperm DNA integrity among the three morphological defects.

Correlation curves between sperm DFI and both clinical features and semen analysis findings are shown in Figure 1. Grade A motility was negatively associated with sperm DFI (r = −0.257; *p* < 0.0001), Grade C and D motility were positively associated with sperm DFI (r = 0.055 and 0.253; *p* = 0.005 and *p* < 0.0001 respectively). Grade B motility is not correlated with sperm DFI (r = 0.032; *p* = 0.0106).

Sperm DFI was positively correlated with age, abstinence time, and semen volume (r values were 0.088, 0.076, and 0.063, respectively; *p* < 0.0001, *p* < 0.0001, and *p* < 0.001, respectively). The sperm concentration, total sperm number, or semen viscosity were found to have no correlation with sperm DNA integrity (*p* > 0.05). The correlation graphs of five major semen parameters are shown in Figure 4.

### 3.3. Multivariate Logistic Regression Analysis of Semen Parameters Associated with Sperm DFI

Despite association analysis illustrating that various semen parameters were positively or inversely related to sperm DNA integrity, some of these clinical variables might be confounding factors. We therefore performed multivariate logistic regression analysis to evaluate the independently determined variables correlated to sperm DFI. We found that age, normal morphology, seminal pH, and progressive motility have a significant overall effect on the DFI. On the other hand, sperm concentration, progressive motility, semen volume, and abstinence days have no significant effect on the DFI after correcting for the former variables.

### 3.4. Subgroup Analysis of Normal Semen Parameters Associated with Sperm DFI

A subgroup analysis of those males who displayed normal semen parameters (in accordance with 5th edition of WHO) was conducted to analyze their sperm DFI. Figure 5 revealed the percentage of cases in each group with their mean of DFI. It was shown that 82.6% of them have DFI of ≤15%, 15.6% of them have DFI between 15−29.9%, and 3% of them have DFI ≥ 30%.

## 4. Discussion

In ART, it has been shown that couples who have unexplained infertility perform less well than those with recognizable underlying causes of infertility. Furthermore, a significant number of the male partners have demonstrated high sperm DNA damage despite having a normal semen analysis profile [29,30]. In a retrospective study from Sweden where semen was analyzed for DFI by SCSA on 127 men from infertile couples with no known female factor and 137 men with proven fertility, it was found that 10.5% of men with proven fertility had a DFI level of at least 20%. On the other hand, a significant proportion of men diagnosed with unexplained infertility according to conventional semen analysis have remarkably high degrees of fragmented sperm DNA [29]. Therefore, impairment of sperm DNA integrity can at least partly explain the subfertility problem of the couple in “unexplained” cases.

In comparison, the results from a French Study applying the Tunel test which undertook a prospective analysis of 1633 patients [31] and a study from Shenzhen in China on another large cohort of 1790 patients’ applying a retrospective analysis of the SCSA Test for correlation between DNA damage and sperm parameters [32]. The majority of their findings were consistent with our study, i.e., both progressive motility and normal morphology were inversely correlated to the rate of DNA damage, with no correlation being found between the DFI and sperm concentration; age, tail defects, and abstinence time were positively correlated with DFI. Another study from Iran, applying both the Halosperm test as well as hypo-osmolar swelling test (HOST), also revealed that there was a significant negative correlation between sperm morphology and DFI [33].

The strength of our study is that we have a large sample size from multi-ethnic males, including Asian and Caucasian. The semen sample for conventional semen analysis and Halosperm test were obtained from the sample that being produced on the same day, albeit the Halosperm reading was deferred from one day to three weeks after collection, while semen analysis was performed on the day of collection immediately after liquefaction. Our study limitation is that this was a retrospective study and data were reliant on the accuracy of record keeping and data entry. We believe this to be highly accurate because of inbuilt validation processes.

Sperm DNA and chromatin defects revealed that sperm DNA damage were associated with an increased risk of pregnancy loss and lower natural IUI and IVF pregnancy rates [16]. Therefore, by knowing that SDF may facilitate decisions of ART modalities by the clinicians, there is a role of instituting antioxidants to high SDF males. Various antioxidant therapies given empirically have been found to be beneficial for infertile men [27,34,35], and a recent meta-analysis also supported those findings, along with the finding that antioxidant therapy improved various indices of male fertility [36]. In addition, lifestyle modifications, including cessation of smoking, exercise in moderation, maintaining a healthy diet, and ideal body mass index, prompt treatment of testicular inflammation, genital tract infections, and varicocele corrective interventions, which may each benefit such cases.

## 5. Conclusions

In conclusion, the quality of spermatozoa is known to play a vital role in fertilization and optimal embryonic development as well as an essential contribution to the generation of healthy offspring. However, conventional sperm parameters are not all clearly correlated with sperm DNA fragmentation integrity. Hence, a sperm DNA fragmentation assay should be considered as an adjunctive role in the investigation of male fertility and to assist with the question of using ICSI in ART cases with unexplained infertility. This study will now extend into a clinical evaluation of the relevance of the DFI of these cases to outcomes from IVF ± ICSI treatments, which are to be reported separately.

## Figures and Tables

**Figure 1 jpm-13-00518-f001:**
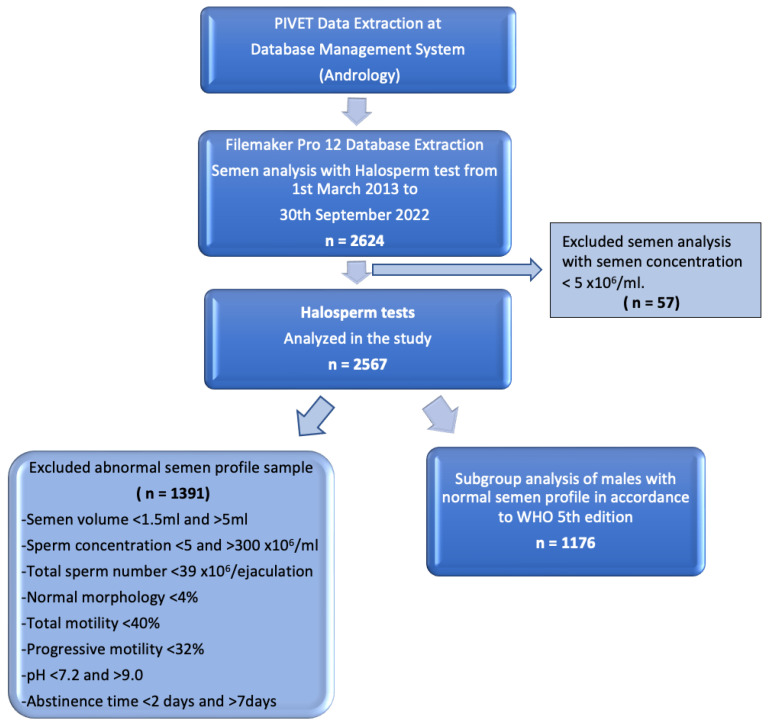
Flow diagram of data extraction displaying groups for Halosperm analysis.

**Figure 2 jpm-13-00518-f002:**
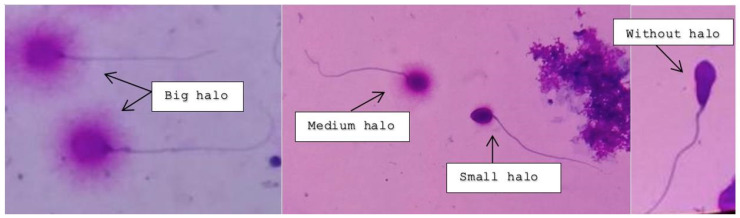
Big and medium-sized halos signify no DNA fragmentation; small and absent halos indicate various rates of DNA fragmentation.

**Figure 3 jpm-13-00518-f003:**
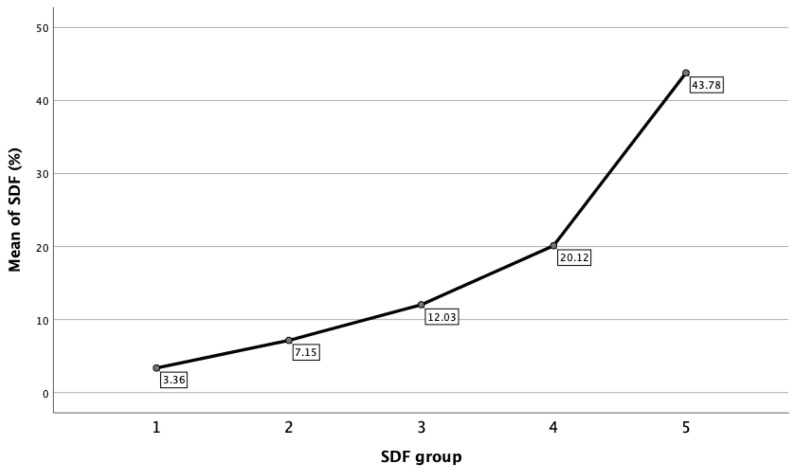
Mean of DFI in each SDF group.

**Figure 4 jpm-13-00518-f004:**
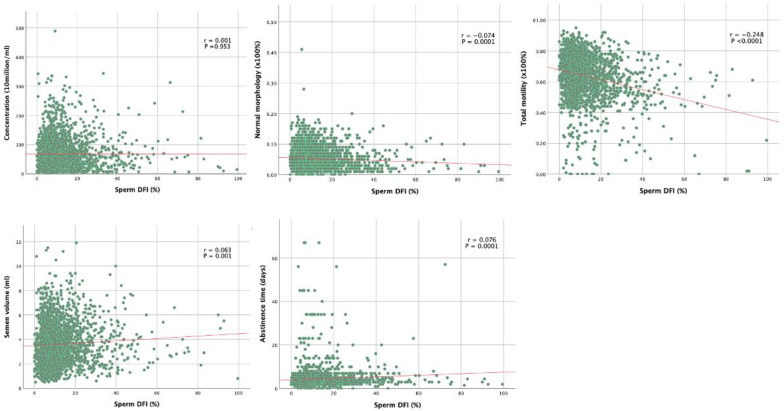
Correlation curves between sperm DFI and both clinical features and semen analysis findings.

**Figure 5 jpm-13-00518-f005:**
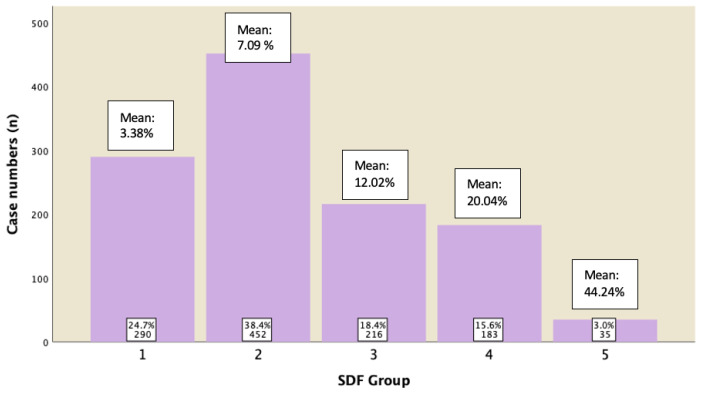
Subgroup analysis for cases displaying normal semen parameters and their DFI ratings.

**Table 1 jpm-13-00518-t001:** Comparison of Mean ± Standard Deviation (SD) and 95% Confidence Interval (CI) in each SDF group.

Variable	1 (*n* = 548) Mean ± SD (95% CI)	2 (*n* = 933) Mean ± SD (95% CI)	SDF Group 3 (*n* = 477) Mean ± SD (95% CI)	4 (*n* = 470) Mean ± SD (95% CI)	5 (*n* = 139) Mean ± SD (95% CI)	*p*-Value
**Sperm concentration (10^6^/mL)**	66.31 ± 44.80 (62.55 − 70.07)	68.35 ± 52.65 (64.97 − 71.73)	69.45 ± 55.08 (64.50 − 74.41)	64.21 ± 50.59 (59.62 − 68.79)	66.98 ± 60.53 (56.82 − 77.13)	0.540
**Normal morphology (%)**	5.47 ± 3.00 (5.22 − 5.72)	5.40 ± 3.26 (5.19 − 5.61)	5.50 ± 3.11 (5.22 − 5.78)	5.01 ± 3.26 (4.72 − 5.31)	4.36 ± 3.26 (3.81 − 4.91)	**<0.001**
**Head defects (%)**	81.22 ± 9.18 (80.45 − 81.99)	81.02 ±10.69% (80.33 − 81.70)	82.65 ± 35.05 (79.49 − 85.80)	81.00 ± 13.01 (79.82 − 82.18)	82.51 ± 8.31 (81.12 − 83.90)	0.473
**Midpiece defects (%)**	9.60 ± 4.70 (9.21 − 10.00)	9.40 ± 4.92% (9.08 − 9.72)	9.06 ± 5.91 (8.52 − 9.59)	8.77 ± 5.40 (8.28 − 9.26)	8.74 ± 5.55 (7.81 − 9.67)	0.053
**Tail defects (%)**	3.15 ± 4.60 (2.77 − 3.54)	3.13 ± 3.01% (2.94 − 3.33)	3.43 ± 3.81 (3.09 − 3.78)	3.35 ± 3.41 (3.04 − 3.66)	4.38 ± 6.63 (3.27 − 5.49)	**0.007**
**Total motility (%)**	66.37 ± 12.33 (65.34 − 67.41)	65.43 ± 12.23% (64.64 − 66.21)	64.60 ± 12.45 (63.48 − 65.72)	60.97 ± 13.92 (59.71 − 62.23)	52.32 ± 17.54 (49.38 − 55.26)	**<0.0001**
**Progressive motility (%)**	61.54 ± 12.93 (60.46 − 62.63)	60.32 ± 12.81% (59.49 − 61.14)	58.94 ± 12.91 (57.78 − 60.10)	55.38 ± 14.11 (54.10 − 56.65)	46.42 ± 17.36 (43.51 − 49.33)	**<0.0001**
**Grade A motility (%)**	39.62 ± 14.55 (38.40 − 40.84)	36.11 ± 15.65% (35.10 − 37.12)	34.39 ± 14.77 (33.06 − 35.72)	29.43 ± 14.68 (28.10 − 30.76)	23.84 ± 14.39 (21.43 − 26.25)	**<0.0001**
**Grade B motility (%)**	21.92 ± 10.72 (21.02 − 22.82)	24.21 ± 13.01% (23.37 − 25.04)	24.55 ± 12.24 (23.44 − 25.65)	25.95 ± 13.36 (24.74 − 27.16)	22.58 ± 13.12 (20.37 − 24.78)	**<0.0001**
**Grade C motility (%)**	4.83 ± 4.63 (4.44 − 5.22)	5.11 ± 4.04% (4.85 − 5.37)	5.66 ± 4.61 (5.24 − 6.07)	5.59 ± 4.17 (5.21 − 5.97)	5.90 ± 5.48 (4.98 − 6.82)	**0.003**
**Grade D motility (%)**	33.62 ± 12.35 (32.58 − 34.65)	34.22 ± 11.83% (33.46 − 4.98)	34.97 ± 11.94 (33.90 − 36.04)	38.60 ± 13.58 (37.37 − 39.83)	47.65 ± 17.56 (44.71 − 50.60)	**<0.0001**
**DFI (%)**	3.36 ± 1.16 (3.26 − 3.46)	7.15 ± 1.38 (7.06 − 7.24)	12.03 ± 1.37 (11.91 − 12.15)	20.12 ± 4.04 (19.76 − 20.49)	43.78 ± 14.36 (41.38 − 46.19)	**<0.0001**
**Age (years)**	35.48 ± 6.06 (34.97 − 35.99)	36.30 ± 6.38 (35.89 − 36.71)	36.25 ± 6.06 (35.71 − 6.80)	36.95 ± 6.78 (36.33 − 37.56)	38.12 ± 7.30 (36.89 − 39.34)	**<0.0001**
**Abstinence time (days)**	3.70 ± 3.64 (3.40 − 4.01)	4.34 ± 4.93 (4.02 − 4.65)	4.96 ± 5.61 (4.45 − 5.46)	4.79 ± 4.62 (4.38 − 5.21)	5.02 ± 5.47 (4.11 − 5.94)	**0.0001**
**Semen volume (mL)**	3.35 ± 1.58 (3.22 − 3.49)	3.54 ± 1.56 (3.44 − 3.64)	3.70 ± 1.61 (3.55 − 3.84)	3.81 ± 1.72 (3.66 − 3.97)	3.64 ± 1.82 (3.33 − 3.95)	**<0.0001**
**Semen pH**	8.08 ± 0.25 (8.06 − 8.10)	8.07 ± 0.27 (8.05 − 8.09)	8.06 ± 0.28 (8.03 − 8.08)	8.04 ± 0.28 (8.01 − 8.06)	8.00 ± 0.31 (7.95 − 8.06)	**0.009**

**Table 2 jpm-13-00518-t002:** Correlations among semen, clinical parameters, and DFI.

Semen Parameters		
Sperm DFI		
	*p*-Value	Correlation (r)
Concentration (10^6^/mL)	0.001	0.953
Total sperm number (10^6^/Ejaculate)	0.019	0.345
Normal morphology (%) Head defects (%) Midpiece defects (%) Tail defects (%)	−0.074 0.009 −0.057 0.096	**0.0001** 0.645 **0.004** **<0.0001**
Total motility (%) Progressive motility (%) Motility Grade A (%) Motility Grade B (%) Motility Grade C (%) Motility Grade D (%)	−0.248 −0.257 −0.254 0.032 0.055 0.253	**<0.0001** **<0.0001** **<0.0001** 0.106 **0.005** **<0.0001**
Age	0.088	**<0.0001**
Abstinence time (Days)	0.076	**0.0001**
Volume (ml)	0.063	**0.001**
PH	−0.066	**0.001**
Viscosity	−0.025	0.214

## Data Availability

Under the terms for Accreditation, the data in this study are not placed in a publicly accessible repository but can be sourced by by correspondence with the Medical Director; corresponding author.

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
