# Peer review of "How Well Do Semen Analysis Parameters Correlate with Sperm DNA Fragmentation? A Retrospective Study from 2567 Semen Samples Analyzed by the Halosperm Test"

_jpm, 2023, doi:10.3390/jpm13030518_

Round 1

Reviewer 1 Report

I would like to thank you for giving me the opportunity to review the article “How well do semen parameters correlate with sperm fragmentation? A retrospective study from 2567 semen samples analyzed by the Halosperm test (second study)”. 

I have read manuscript with keen interest. It is a retrospective comparative analysis of sperm parameters and sperm DNA fragmentation on a large data set collected over 10 years in a single ART facility. Authors reveal that sperm DNA fragmentation index was significantly correlated with age as well as sperm motility and morphology and suggest that sperm DNA fragmentation assay should be included in the evaluation steps of male infertility cases. 

The manuscript is well written. The purpose of the study is clear. Study population, methods related with conventional sperm analysis and sperm DNA fragmentation analysis are well-described. Statistical analysis is appropriate and results are well- presented and discussed. 

I have a few minor suggestions:

1.    I could not figure out why the authors included “(second study)” in the Title.

2.    Reference no 14 is misplaced in Line 69.

3.    Figure-1, particularly from analysis to subgroup analysis, seems kind of puzzling. Is it possible to modify for ease of understanding?

4.    Is the presumption given in lines 282-283 necessary?

5.    Please double check the correct way to cite WHO Lab Manuals in References no 7 (line 410) and 14 (line 424).

Author Response

Reviewer #1.

  1. This manuscript is one of three associations covering the Halosperm Test – the first study covering the male clinical aspects; the second concerning the semen analysis profile; and the third study detailing the outcomes from assisted reproduction treatments. However, we would now believe it to be unnecessary to include reference to Study two in the Title, hence deleted.
  2. Reference no 14 is indeed misplaced in line 69; it has already been cited in line 66 and has therefore been removed.
  3. Figure 1 (The Flow Chart) has been extensively revised for clarification, mainly to delineate the males with a normospermic profile from those with an abnormal semen analysis; the former having a separate sub-group analysis.
  4. We agree that the presumption on lines 282-283 (Results) is inappropriate. The sentence has been removed.
  5. WHO citation corrected (along with all references according to ACS style).

Reviewer 2 Report

The current study aims to investigate the association between sperm quality assessed by conventional semen analysis and sperm DNA integrity assay applying the Halosperm test.

The study demonstrated that a sperm DNA fragmentation assay should be considered as an additional step in the investigation of male fertility and to assist with the question of using ICSI in ART cases with unexplained infertility.

The authors should be congratulated for their work and for addressing an important topic. Only few points warrant mentions:

Minor comments:

1.    In the “Material and Methods” section, I suggest to show also adopted thresholds for each sperm parameter.

2.    in the “Discussion” section, the authors should also discuss the potential role of clinical parameters as predictors of finding altered SDF in men seeking for medical help for infertility, such as in PMID: 36598012.

Author Response

Reviewer #2.

  1. The adopted thresholds for each sperm parameter have now been detailed within the Material and Methods section.
  2. We thank the Reviewer for drawing our attention to article PMID:36598012 which has only recently been published. We have now expanded our Discussion to consider the implications of the Halosperm test result vs the semen analysis profile. This will be better analysed in our impending first study detailing the association of Halosperm test with male clinical features.